# GDBM: A database of global drainage basin morphology

**Stuart W. D. Grieve**[1,2]*, **Shiuan-An Chen**[7],
**Michael B. Singer**[3,4,5], **Katerina Michaelides**[5,6]

**1** School of Geography, Queen Mary University of London, London, United Kingdom, **2** Digital Environment Research Institute, Queen Mary University of London, London, United Kingdom, **3** School of Earth and Environmental Sciences, Cardiff University, Cardiff, United Kingdom, **4** Water Research Institute, Cardiff University, Cardiff, United Kingdom, **5** Earth Research Institute, University of California Santa Barbara, Santa Barbara, California, United States of America, **6** School of Geographical Sciences, University of Bristol, Bristol, United Kingdom, **7** Institute of Water Resources Development, Feng Chia University, Taichung, Taiwan

\* s.grieve@qmul.ac.uk

## Abstract

Rivers and their drainage basins are fundamental landscape units, and their morphology is a record of the cascade of geologic, tectonic, biological, and climatic processes acting upon them. Quantifying this cascade depends on morphometric measurements of rivers and drainage basins, and comparison of these measurements across diverse landscape settings. Here we present a new near-Global dataset of Drainage Basin Morphology, GDBM, which provides morphometric measurements of 254,966 basins and the longest river channel within them. This dataset is created by extracting channels from the 30-meter resolution Shuttle Radar Topography Mission (SRTM) topographic data which fall within Köppen-Geiger climate zones, to allow the influence of climate on river and basin morphology to be quantified. GDBM contains measurements of channel length, slope, relief, normalised concavity, basin area, basin shape and aridity. These data have been generated with minimal assumptions, focusing on identifying and classifying channels with high confidence, through the use of a conservative drainage area threshold. GDBM provides opportunities for rapid spatial analysis of channel morphology at a near-global scale and has the potential to yield continuing insight into landscape evolution across diverse climate regimes. This dataset also has potential applications across a range of Earth and environmental science domains, through the integration of additional data on, for example, forest canopy height, landcover, or soil properties to explore the spatial variability of channel and basin properties with climate.

## 1 Introduction

Rivers and their drainage basins are critical components of landscapes which exist at spatial scales spanning several orders of magnitude, driving the distribution of water and sediment across the Earth's surface [1]. They exist across several orders of magnitude of spatial scales, from continental river systems through to hillslope scale drainage networks [2,3].

**Data availability statement:** The dataset described in this manuscript can be accessed at https://doi.org/10.5281/zenodo.7970485

**Funding:** The author(s) received no specific funding for this work.

**Competing interests:** The authors have declared that no competing interests exist.

The analysis of channel and basin morphometry has yielded a diverse range of insights into landscape response to tectonic [4–7], climate [8–11] and anthropogenic [12] forcing, in addition to informing the parameterization of hydrologic [13,14] and landscape evolution [15–17] models.

With the increasing availability of high quality global and near-global topographic data, a number of compilations of global channel data have been produced, notably MERIT Hydro [18], HydroSHEDS [19], HDMA [20], and Basin90m [21]. These datasets are valuable resources for many avenues of research, typically however this existing family of datasets are not directly suited to analysis which links climate and channel morphometry. For some datasets, river mapping approaches make use of published blue line maps [22], which perform well in humid environments but systematically exclude intermittent and ephemeral channels, particularly those found in drylands [23,24]. In the case of Basin90m [21], rivers in drylands are removed based on their aridity, excluding many ephemeral channels from analysis. In other cases, morphometric data is extracted using HydroSHEDS as the input for basin extraction [25], again potentially excluding intermittent and ephemeral channels from the dataset. Some datasets make use of a stream burning approach [26] to enforce channelised flow across a DEM and this approach ensures a topologically consistent network, but modifies the topographic data, potentially biasing measurements of channel or basin morphology. Other datasets are predicated on lower resolution DEM products [19,21,27], reducing their ability to identify smaller channels. Several datasets do not record geomorphometric data natively [26], requiring a user to load the channel data, source a DEM and then sample the required morphometric data. This lack of native morphometric information can lead to inconsistency and a reduction in the spatial scope of analyses. With the aim of addressing these limitations, supplementing existing datasets and facilitating the analysis of relationships between climate and basin morphometry, we present a new near-Global dataset of Drainage Basin Morphology, GDBM.

Identifying the initiation point of channels is very challenging, such that successful schemes are only feasible on small spatial scales or with large amounts of manual intervention [28]. Rather than developing new methods to delineate channel initiation points, a challenging research topic known to be limited by data resolution [29], we instead focus here on extracting the longest channel from large drainage basins, with a parsimonious drainage threshold [30] of 22.5 km$^2$). In doing this we can be confident that the channels we extract within GDBM are not false positives, supporting global analysis of channel morphology undertaken without undue influence from potential channel extraction biases. The power of this new dataset is its minimal, parsimonious assumptions and its close coupling to Köppen-Geiger climate zones [31] and global aridity estimates [32,33]. GDBM thus creates opportunities to explore river and basin morphology in a climate context at a near-global scale, from the profiles of individual channels, to continental scale statistical properties.

## 2 Materials and methods

The processing and generation of GDBM follows a series of stages:

### 2.1 Climate zone processing

A fundamental component of GDBM is the connection of climate categories and topographic data at an appropriate scale for near-global analysis. We use the Köppen-Geiger climate classification [31] to divide the globe into discrete climate sub-zone tiles, which can be processed in parallel. As the creation of this dataset is motivated by an interest in climate-drainage basin

relationships at a near-global scale, the full climate classification extended by [31] is too granular. To this end, we generalise the climate sub-zones into the categories described in Table 1, and exclude the **ET** and **EF** polar classifications. In some cases the sub-zones are still too big to be processed efficiently. These are further divided using a quadtree-like algorithm to maintain uniform tile shapes (Fig 1a). This division of the Earth's surface into a series of large tiles which can be processed in parallel is in line with other efforts to create large datasets of channel morphology [21] and represents a compromise between computational power and dataset scale.

Following the creation of the climate sub-zone tiles, large bodies of water are removed from each tile, to ensure that the tiles only represent terrestrial environments. We used the Global Lakes and Wetlands Database [34] for this purpose. This dataset characterises all classified bodies of water into two levels: level 1 is made up of all bodies of water with a surface area above 50 km$^2$ and level 2 represents all remaining bodies above 0.1 km$^2$. Consequently, all level 1 lake and reservoir polygons were intersected with the climate sub-zone tiles using shapely [35]. In rare cases the resultant intersection between water bodies and climate sub-zones created multiple polygons (for example if a lake bisected a climate sub-zone tile). Therefore if a resultant split polygon had an area less than 20% of its original size, it was classed as a sliver and removed from further analysis [36].

## 2.2 Topographic data processing

The river channels provided within this dataset are extracted from the NASA Shuttle Radar Topography Mission Global 1 arc second DEM product [37,38], henceforth referred to as SRTM. This is the latest version of the SRTM dataset which has undergone extensive void-filling and quality control. The SRTM topographic data is used extensively in global analyses of topography [10,39], and lower resolution versions of the dataset have underpinned previous efforts to create global hydrologic datasets [18,19].

Due to the nature of the space shuttle's orbit, topographic data was only collected between 60°N and 56°S [37]. This limits the creation of a truly global dataset, however, as the purpose of these data are to understand the relationships between climate and fluvial channel morphometry, excluding polar data where channel forms will be carved by ice is beneficial. The latest version of the SRTM data has a grid resolution of approximately 30 m at the equator

**Table 1. Details of aggregated Köppen-Geiger climate zones used in this dataset, and short descriptions of each of these zones. Note that polar zones ET and EF are excluded from this dataset.**

| Letter Code | Description | Original Codes |
|---|---|---|
| Af | Tropical-Rainforest | Af |
| Am | Tropical-Monsoon | Am |
| Aw | Tropical-Savannah | Aw |
| BWh | Arid-Desert-Hot | BWh |
| BWk | Arid-Desert-Cold | BWk |
| BSh | Arid-Steppe-Hot | BSh |
| BSk | Arid-Steppe-Cold | BSk |
| Cs | Temperate-Dry summer | Csa, Csb |
| Cw | Temperate-Dry winter | Cwa, Cwb, Cwc |
| Cf | Temperate-Without dry season | Cfa, Cfb, Cfc |
| Ds | Cold-Dry summer | Dsa, Dsb, Dsc, Dsd |
| Dw | Cold-Dry Winter | Dwa, Dwb, Dwc, Dwd |
| Df | Cold-Without dry season | Dfa, Dfb, Dfc, Dfd |

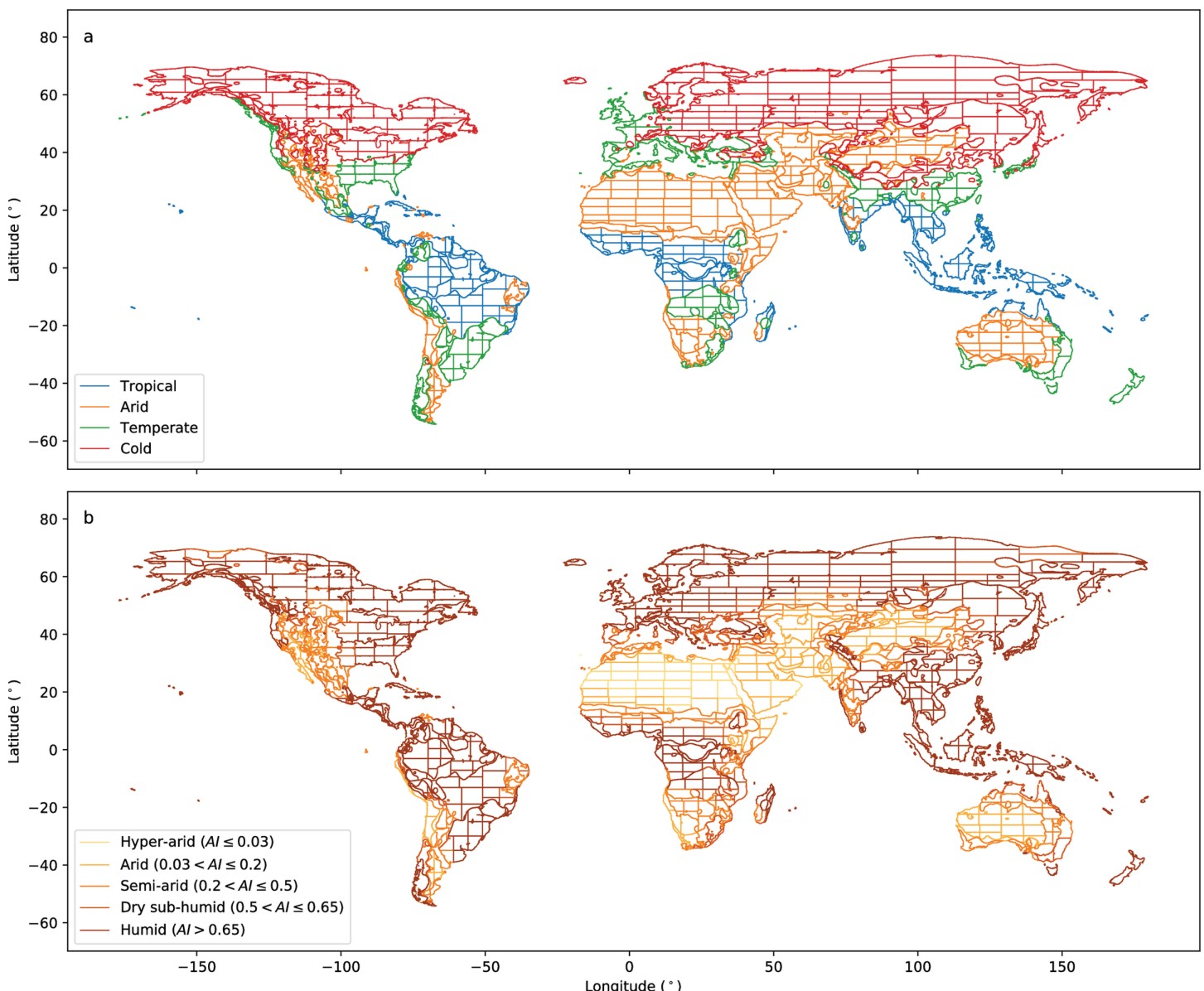

**Fig 1. Maps of the 1830 climate sub-zone tile boundaries.** Classified by a) Köppen-Geiger climate zone and b) Aridity Index. Note that channels within each tile will have a range of AI values, as AI is recorded on a per channel basis within GDBM.

[38], providing a good balance between computational efficiency and data fidelity. For a full discussion of SRTM data quality and validation, see [40].

Topographic data for a given climate sub-zone was downloaded using the OpenTopography Service [41], allowing the relevant data to be downloaded and processed on the fly, rather than requiring the whole SRTM dataset to be downloaded or processed at once. Using GDAL [42], the SRTM tiles which intersect with a given climate sub-zone are then merged into a single DEM tile, and this merged tile is clipped to the polygon outline of the climate sub-zone to

create a DEM which covers only the climate sub-zone of interest. To facilitate accurate comparisons between climate sub-zones, each tile is then reprojected into the appropriate UTM zone, based on the location of the southwest corner of the climate sub-zone.

The final stage of topographic processing is to hydrologically correct each clipped and projected DEM. This is required to identify and remove topographic depressions which inhibit surface flow paths and render channel extraction impossible [43]. At the resolution of the SRTM data, and following the clipping out of lakes and water bodies, the majority of depressions will be data artefacts rather than true topographic features [44]. However, GDBM includes metrics which can be used when analysing these data to control for channels unduly impacted by erroneous hydrological correction. This processing stage is performed using the LSDTopotools [45] implementation of the Wang and Liu algorithm [46]. This algorithm applies hydrological corrections to topographic data in a computationally efficient manner, and has been shown to be robust under a number of geomorphic applications [5,47–49]. Other hydrological correction algorithms have been developed, notably those that consider hydrological context alongside high resolution topographic data [50–52]. However, given the resolution of the SRTM data being processed, and the scale over which this dataset is being generated, increasing the complexity of the hydrological correction algorithm would yield few benefits, weighted against the considerable additional computational cost.

The final result of these processing steps is the generation of 1830 hydrologically corrected and projected DEM tiles, corresponding to each of the climate sub-zone tiles generated previously.

## 2.3 Channel extraction

Channel identification and extraction from topographic data is a common problem in geomorphology [53]. Since the widespread adoption of LiDAR topographic data, a range of algorithms have been developed, either attempting to identify a process domain boundary where fluvial processes outcompete hillslope processes [54] or attempting to identify a geomorphometric signature of channelisation [55–58]. These methods have been demonstrated to be effective when applied to high resolution topographic data, but have limited efficacy at SRTM's 30-meter resolution [29]. Consequently a more conservative channel extraction approach is employed using a drainage area threshold to identify the initiation point of each channel [30,59,60]. Drainage area is computed using the Fastscape implementation of the D8 steepest descent algorithm [61], which is designed to work optimally over large spatial scales.

There is no globally appropriate drainage threshold which can be applied confidently to extract river channels with an absence of both false positives and false negatives [56]. If too low a threshold is chosen, channels will be identified in the data where none exist in reality, and if too large a threshold is chosen, only the largest rivers in a drainage basin will be identified. Here, we exploit this feature of drainage area based channel extraction by using a deliberately conservative fixed threshold of 22.5 km$^2$. This threshold ensures that every channel extracted has a high probability of corresponding to a true channel.

GDBM only records morphometric information about the mainstem channel, defined as the longest channel in a drainage basin, and so the loss of tributary channels due to conservative drainage thresholding does not impact the overall compilation of the dataset. To ensure that each channel within GDBM corresponds to a distinct climate zone, drainage basins which cross or intersect with a climate sub-zone tile boundary are not recorded within the dataset (n=35,979). This filtering of data to exclude boundary crossing channels maximises the value of our dataset as a tool to explore relationships between climate and basin

morphometry. We also test for nesting of drainage basins, to ensure that each channel is only recorded once in the dataset, avoiding problems of serial correlation within the data.

## 2.4 Aridity index processing

In addition to the Köppen-Geiger climate sub-zone data, each river channel in the database records Aridity Index [32,33] (AI) values along its length. This sampling process records AI values at the centroid of every channel pixel, resulting in an average sampling frequency of 36 meters along each channel. These values can be used to explore along channel variability in aridity, within Köppen-Geiger climate sub-zones (Fig 1b).

From this population of sampled values the mean and median AI values for each channel are calculated, in addition to standard deviation, maximum and minimum values. Due to the resolution disparity between the SRTM dataset (∼30 m) and the Aridity Index dataset (∼900 m) some channels have a small number of Aridity Index measurements along their length (<10), but this only accounts for 10 channels, or 0.004% of the whole dataset. The number of individual AI values for each river is therefore also recorded, to allow users to filter out such rivers as required.

## 2.5 Channel and basin morphometric calculations

Following the extraction of channels within each of the climate sub-zone tiles, a series of channel morphometrics are calculated. Channel relief ($R$) is calculated as:

$$R = E_0 - E_n \tag{1}$$

where $E$ is channel elevation and the subscripts 0 and $n$ correspond to the upstream and downstream extent of the channel, respectively. Flow length ($L_f$) is calculated by:

$$L_f = L_n - L_0 \tag{2}$$

where $L$ is the cumulative upstream flow distance at a given point along the channel. Total channel slope (S) is computed by:

$$S = \frac{R}{L_f}. \tag{3}$$

The Normalised Concavity Index (NCI)[10] is calculated by fitting a straight line through the points $E_0$, $E_n$, described by the equation $Y_L = E_0 - \theta L$ where $\theta$ is the gradient of the line, the y intercept is $E_0$ and $Y_L$ is the elevation of the line at position $L$ along the line. NCI can then be calculated at each channel pixel as follows:

$$NCI = median\left(\frac{E_L - Y_L}{R}\right). \tag{4}$$

Similar calculations can be performed at reach scale rather than along the whole channel, using the individual river data which are described below.

The Gravelius compactness coefficient (GC) [62], the ratio between a basin's perimeter and the circumference of a circle of the same area, is used to describe basin shape, with a value of 1 indicating a circular basin and increasing values indicating increasing basin elongation. Perimeter estimation has been shown to be impacted by data resolution and basin size, where

increasing basin size leads to increasing perimeter overestimation [63]. To resolve this issue, we follow [64] in defining a relative resolution ($R_r$):

$$R_r = \frac{1}{10}\sqrt{A} \tag{5}$$

where $A$ is the basin area. This relative resolution can be used to convert basin perimeter ($P$) into relative perimeter:

$$P_r = P \cdot R_r \tag{6}$$

and convert basin area into relative area:

$$A_r = A \cdot R_r^2 \tag{7}$$

and using these relative values, the value of GC can be calculated:

$$GC = \frac{P_r}{2\sqrt{\pi A_r}}. \tag{8}$$

## 2.6 Quality assurance metrics

In addition to collecting topographically derived information about each channel, we also record information that can be used to quality control the dataset. Common concerns when working with topographically defined channels are that the pit filling procedure may distort the *true* data or that the limitations of the D8 algorithm will create anomalously straight channels.

To address the potential impact of the pit filling procedure on channel morphometrics, DEMs of difference are generated between the filled and unfilled DEMs. Fig 2 shows the distribution of topographic change across an example climate sub-zone tile caused by the hydrological correction process. The majority of topographic changes fall well below the SRTM relative vertical error of between 4.7 and 9.8 m [40] and so are excluded from further analysis by filtering the data to the 98th percentile. The remaining data corresponds to all pixels within a climate sub-zone which have been altered by more than the reported vertical error within the data. The proportion of each channel impacted by these altered pixels can then be calculated both in terms of raw pixel counts and flow length to create quality assurance metrics.

We define channel straightness by identifying the longest unbroken run of flow directions within each channel. The length of this run is then compared to the complete channel length to create a straightness quality assurance metric, where a value of 1 would denote a completely straight channel, and values close to 0 denote high variability in channel flow direction.

## 3 Results

### 3.1 Data records

**3.1.1 Aggregate data.** For each climate sub-zone (see Table 1) a `csv` file has been created, which contains summary statistics and geographical information for each channel within that climate sub-zone, with one river corresponding to each row in the dataset. Within GDBM

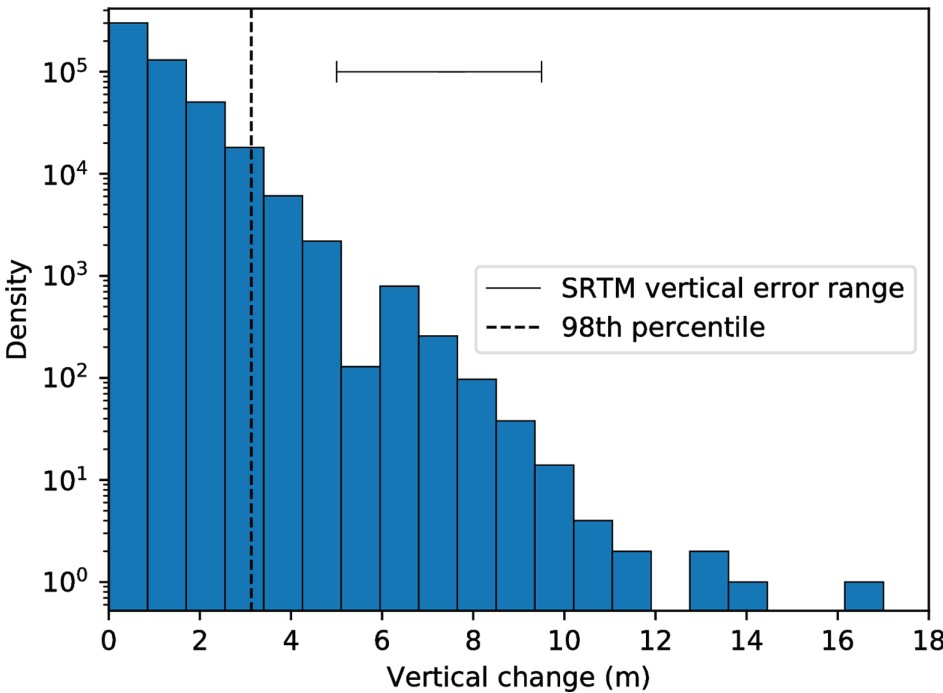

**Fig 2. Histogram of the distribution of per-pixel vertical change following hydrological correction for a representative climate sub-zone tile.** Dashed vertical line indicates the 98th percentile threshold for meaningful vertical change. Error bar shows the range of vertical error within the SRTM dataset [40].

every river is given a unique name, for example `Af_74_9435e317_8045_44ab_bba7_cb9ff452496f_river_46` which is composed of:

- The Köppen-Geiger letter code. **Af**.
- A tile ID number, indicating that this river is found within the nth tile of that climate sub zone. **74**.
- An optional unique ID, used to denote where sub-zone tiles have been further subdivided. **9435e317_8045_44ab_bba7_cb9ff452496f**.
- A river ID, denoting that this is the nth river of that tile. **river_46**.

It is important to note that no topological information is stored within the unique river names, and so it is not possible to assume any spatial relationship between sub-zones based on their numerical value. In total there are 13 csv files, with a total of 254,966 records and the size of this portion of GDBM is 69 MB. Table 2 provides a description of each column within the aggregate data files.

**3.1.2 Individual river data.** Each row in the aggregate data corresponds to a `csv` file, named using the unique name described above, which contains the data for each pixel along the river's length. In total there are 254,966 of these river files, organised into Köppen-Geiger climate sub-zone subdirectories (see Table 1), these river files have a total size of 27 GB. To make the computational processing of large batches of data more efficient, these river files do not have a header row. Table 3 provides a description of each column within these river files, in the order that they appear in the file. The individual river files are structured

**Table 2. Details of the variables recorded within the climate sub zone aggregate data files.**

| Variable name | Description | Units |
|---|---|---|
| RiverName | Unique name for each river within the dataset. | - |
| NCI | Normalised Concavity Index. Calculated using Eq 4, following [10] and detailed in Sect 2.5. | - |
| Relief | Total channel relief, calculated as the difference between the maximum and minimum elevations within the channel, Eq 1. | m |
| FlowLength | The total along channel length of the river. | m |
| TotalSlope | Total channel gradient calculated as the ratio between the relief and the flow length, Eq 3. | m/m |
| Area | The total drainage area for the channel. Computed at the lowest elevation pixel within the channel. | $m^2$ |
| ai_mean | The mean Aridity Index value for the channel. Calculated by sampling all Aridity index values along the channel and calculating their mean. | - |
| ai_median | The median Aridity Index value for the channel. Calculated by sampling all Aridity index values along the channel and calculating their median. | - |
| ai_std | The standard deviation of Aridity Index values for the channel. Calculated by sampling all Aridity index values along the channel and calculating their standard deviation. | - |
| ai_min | The minimum Aridity Index value along the channel. | - |
| ai_max | The maximum Aridity Index value along the channel. | - |
| ai_n | The total count of Aridity Index values sampled along the channel. | - |
| pit_pixel_proportion | The proportion of channel pixels which the hydrological correction process has altered by more than the SRTM vertical error, following the process described in Sect 2.6. | - |
| pit_length_proportion | The proportion of the channel by length which the hydrological correction process has altered by more than the SRTM vertical error, following the process described in Sect 2.6 | - |
| straightness_proportion | The ratio between the length of the longest anomalously straight section of channel and the total channel length, following the process described in Sect 2.6 | - |
| perimiter_pixels | The total number of pixels making up the basin perimeter. | - |
| area_pixels | The basin area in pixels. | - |
| Gravelius_coefficient | The Gravelius compactness coefficient. Calculated using Eq 8, following [64] and detailed in Sect 2.5. | - |

so that each row corresponds to a single pixel within the channel, with the first row corresponding to the outlet of the channel and the last row corresponding to the upper limit of the channel.

## 4 Technical validation

As discussed above, the channels that make up GDBM are extracted using a conservative drainage area threshold to maximise the likelihood that the dataset consists of true channels. We explore the impact that selecting such a parsimonious drainage area threshold may have on the parameters extracted for each channel through a sensitivity analysis. The full channel extraction process was run using drainage area thresholds ±25% of the standard value of 22.5 km². Fig 3 shows the distribution of the NCI statistic for the four broad climate categories for each of the drainage thresholds. From these distributions we can conclude that the choice of drainage threshold does not have a meaningful impact on channel properties within GDBM at these scales.

Fig 4a shows the distribution of the NCI statistic, under differing levels of filtering based on the hydrological correction quality assurance metric. When the dataset only retains channels where less than 0.5% (n=163,383) or 0.1% (n=136,530) of the total channel length have

**Table 3. Details of the variables recorded within each individual river file. Note that the order of rows in this table corresponds to the order of the columns within the dataset.**

| Variable name | Description | Units |
|---|---|---|
| row | Row-wise pixel coordinate within the climate sub-zone tile. | - |
| col | Column-wise pixel coordinate within the climate sub-zone tile. | - |
| latitude | Latitude of a channel pixel recorded using WGS84 datum with EPSG code 4326. Northern hemisphere values are positive and southern hemisphere values are negative. | decimal degrees |
| longitude | Longitude of a channel pixel recorded using WGS84 datum with EPSG code 4326. Eastern hemisphere values are positive and western hemisphere values are negative. | decimal degrees |
| elevation | The elevation above sea level of a channel pixel. | m |
| flow length | The cumulative upstream flow distance of the channel. Note that the first row of the dataset does not equal zero and must be subtracted from all flow length values if comparisons between rivers are to be made. | m |
| drainage area | The cumulative upslope drainage area of a channel pixel. Drainage area values increase with downstream distance. | $m^2$ |
| basin key | An internal LSDTopoTools ID, used here to assign river numbers in each river's unique name. | - |
| flow direction | An integer flag denoting the flow direction of a channel pixel, as computed using the Fastscape algorithm [61]. 0 denotes North, and values increment clockwise, concluding with Northwest denoted by 7. | - |
| aridity index | The sampled Aridity Index [32,33] value for a channel pixel, the sampling process is described in detail in Sect 2.4. | - |
| pit flag | A boolean variable set to 1 if a pixel has been modified in excess of the vertical error in the SRTM data by the hydrological correction and 0 if it remains unmodified. | - |
| perimeter pixel count | The number of pixels that make up the basin perimeter. | - |
| area pixel count | The number of pixels that make up the basin area. | - |
| easting | The easting of the channel pixel in UTM. | m |
| northing | The northing of the channel pixel in UTM. | m |

been impacted by the hydrological correction process, there is limited variation in the distribution of NCI values, even though there is a large reduction in the number of channels that pass these threshold when compared to the unfiltered dataset.

A similar process can be applied to explore how anomalous channel straightness may impact the dataset (Fig 4b). In this case, there is a large reduction in the median values when channels with greater than 0.1% of their total length being flagged as straight are excluded. For most applications, this will be an overly prescriptive level of filtering, as for the median length river in the dataset (31,897 m), this corresponds to a straight length of approximately 32m, or less than 2 pixels. However, using a more appropriate threshold of 0.5% preserves the same distribution and median as the unfiltered dataset.

A frequently espoused limitation of topographically defined channels, particularly those extracted from global topographic datasets, is their inability to capture channel sinuosity. We explore the ability of GDBM to capture large scale channel sinuosity, by segmenting each channel into 10 km long reaches and calculating the ratio of channel flow length to the straight line distance between the start and end point of each reach. If this sinuosity ratio is greater than 1, the flow length of a reach is longer than the Euclidean distance, indicating a meandering channel planform. In cases where the sinuosity ratio is less than 1, the channel is meandering at much longer wavelengths. Fig 5 demonstrates the relationship between these two distance metrics, and that GDBM is indeed capturing channel sinuosity at an appropriate scale.

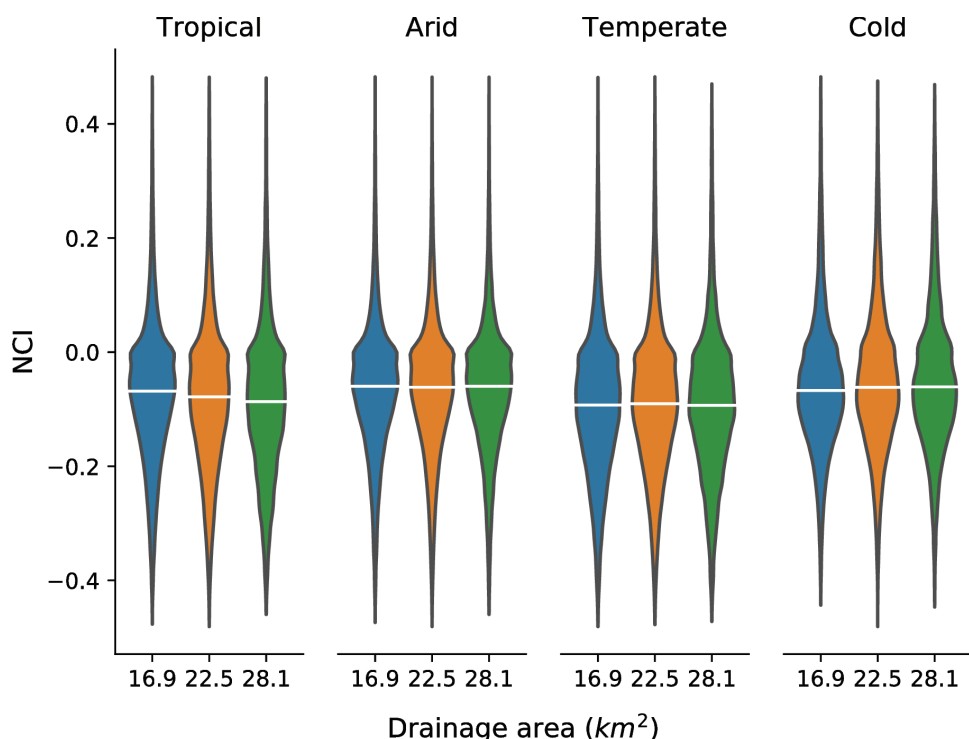

**Fig 3. Violin plots of NCI values calculated for every channel within GDBM, grouped into broad Köppen-Geiger climate zones.** For each climate category, three channel initiation thresholds have been used to explore the influence of this parameter on the consistency of the dataset as a whole. White bar on each violin corresponds to the median value.

The validity of extracted channels can also be assessed through direct comparison with existing global channel network datasets. Fig 6 shows example GDBM channels from the four broad climate zones, alongside HydroSHEDS [19] channels from the same geographic region. The motivation of these comparisons is not to critique existing datasets, but rather to demonstrate the validity of GDBM and to draw distinctions between the datasets and their different use cases. By design, GDBM channels are sparser than the HydroSHEDS channels due to the parsimonious design of GDBM to only extract the longest channel within each basin, using a conservative drainage area threshold. Across all four climate zones, the GDBM data shows broad agreement with the HydroSHEDS channels, however in the lower relief arid example (Fig 6b) the HydroSHEDS channels do not follow the higher resolution GDBM channel, highlighting the challenges of extracting dense networks in such terrain. When comparing a GDBM channel to its HydroSHEDS counterpart, we generally observe a more sinuous channel planform, conforming to the landscape morphology. This improved representation of sinuosity is a function of the resolution of the topographic data used to create GDBM, and provides us with confidence that the channel profile data contained within GDBM is a reasonable representation of the true channel morphology at these spatial scales.

## 5 Usage notes

Version 1.0.0 of the GDBM dataset has been uploaded to the Zenodo data repository [65], this is the canonical location to access this and any future versions of the dataset. All of the GDBM data is in a `csv` format, designed for maximum interoperability between different

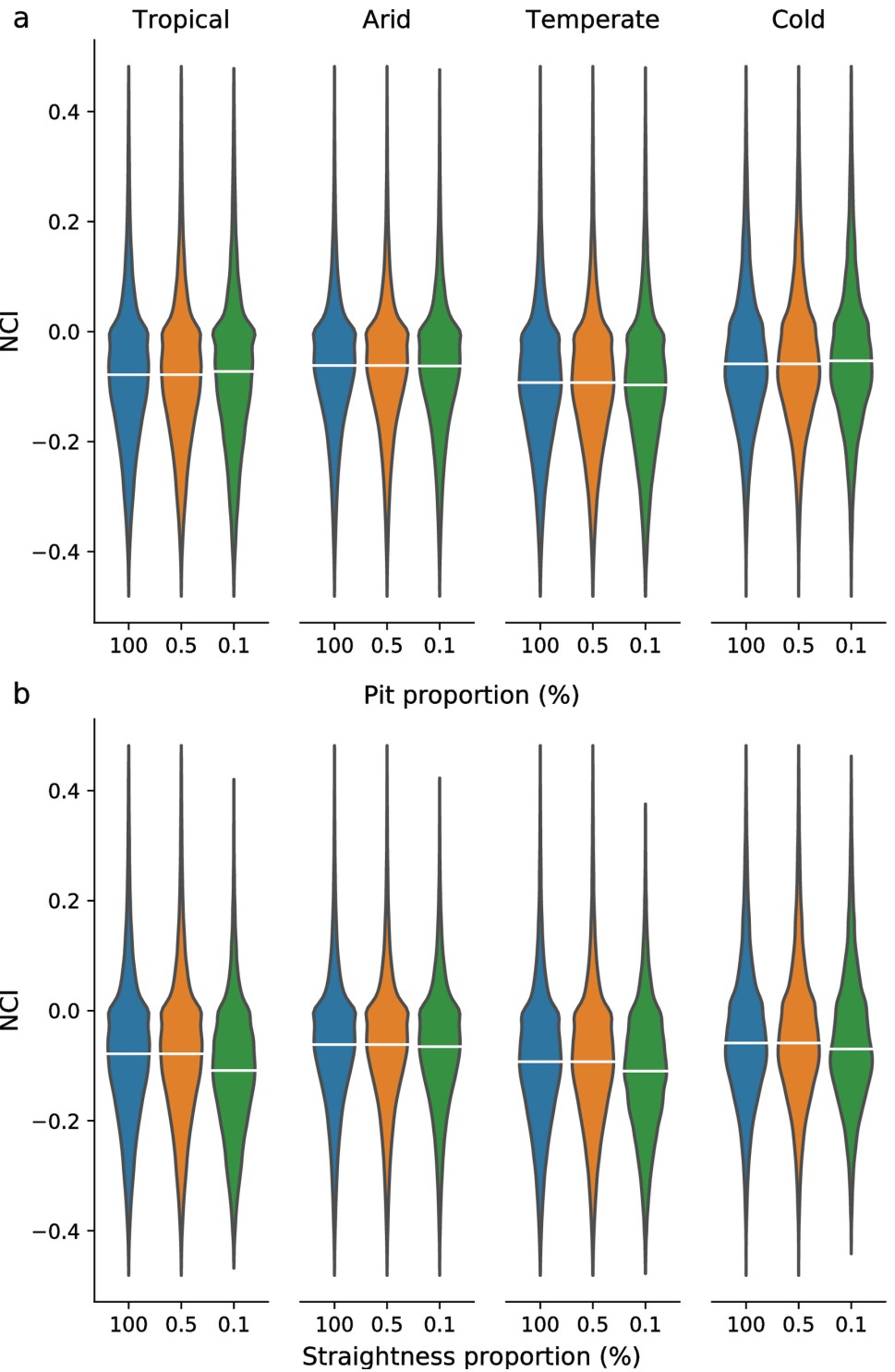

**Fig 4. Violin plots of NCI values calculated for every channel within GDBM, grouped into broad Köppen-Geiger climate zones.** For each climate category, three levels of filtering using quality control metrics are used, to explore the potential impact of the channel extraction process on measures of channel morphology. Data is filtered to exclude channels which have a quality control metric exceeding the reported value. A value of 100 indicates no filtering has been applied. a) number of hydrologically corrected pixels as a proportion of total channel length. b) length of longest straight channel segment as a proportion of total channel length. White bar on each violin corresponds to the median value.

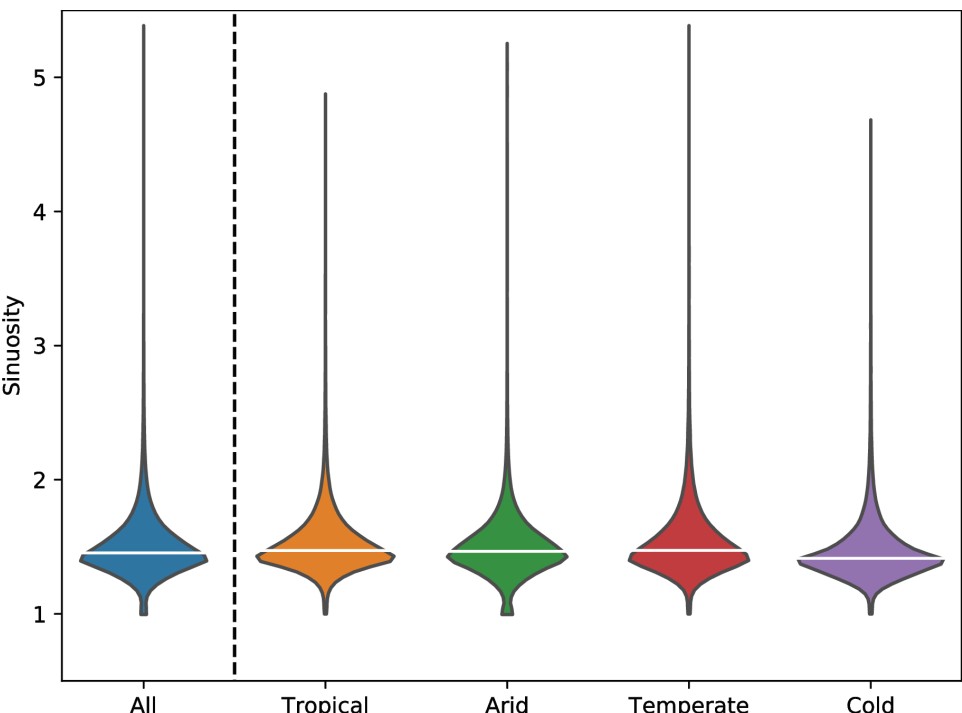

**Fig 5. Violin plots of the sinuosity ratios calculated for every 10 km reach within the dataset.** Sinuosity data is grouped into broad Köppen-Geiger climate zones, as well as aggregated into a single dataset. Extreme outliers accounting for approximately 0.01% of the data has been excluded to enhance clarity of the plot. In each case the majority of reaches demonstrate a sinuosity ratio above 1, corresponding to the identification of meandering channel planforms within GDBM. White bar on each violin denotes the median value.

analysis environments. Due to the volume of data it is expected that users of the dataset will interrogate GDBM programatically. The authors recommend the use of NumPy [66] and pandas [67,68] to read and analyse the data, or their equivalents in other programming languages. Individual channel planforms can be visualised inside any modern GIS package, for example QGIS [69].

Code to generate all of these data, end to end (including full documentation), will be archived alongside this paper (prior to publication, this code can be accessed at https://github.com/sgrieve/gdbm). The code has been developed to run on the QMUL Apocrita HPC facility [70], which runs Univa Grid Engine, and so job scripts, file paths and virtual environments will need to be adapted to run the code on other HPC systems. However, the actual data processing code will not need to be modified between systems. The authors note that there is a considerable energy cost to running code on HPC systems [71], with the generation of this dataset estimated to consume 188 kg $CO_2$ equivalent [72]. However, in most use cases, there should be no need to re-generate the GDBM data from scratch, lowering the overall environmental cost of this research. Avoiding data re-processing through data sharing is an important component of lowering the climate impact of computational research. Alongside the code to generate the GDBM data, code to generate the figures in this paper is included, which acts as additional documentation of how these data can be analysed using Python.

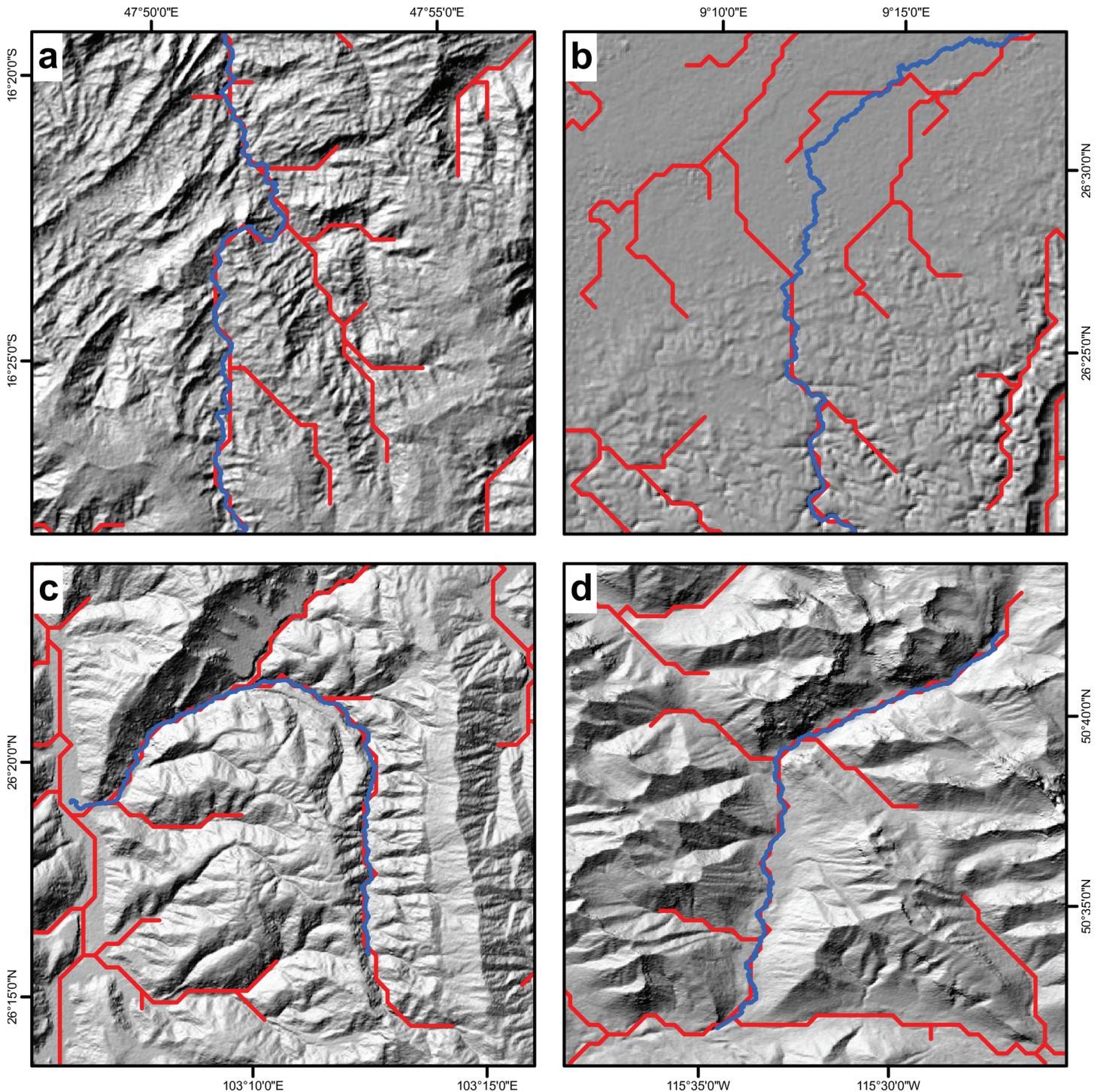

**Fig 6. Maps showing comparisons between GDBM (blue) and HydroSHEDS [19] (red) channels across: a) tropical; b) arid; c) temperate; and d) cold Köppen-Geiger climate zones. Underlying hillshade is generated from the NASA Shuttle Radar Topography Mission Global 1 arc second DEM product [37,38].** Coordinates are in the WGS84 datum with EPSG code 4326.

## 6 Code availability

All code used in the generation and processing of GDBM, in addition to the code used to generate the figures in this paper have been released under the open source MIT Licence and are available online (https://github.com/sgrieve/gdbm). All parameters used within the processing workflow are documented within the code repository.

## Acknowledgments

The authors thank the creators of the datasets used in this study for making them publicly available. This research utilised Queen Mary's Apocrita HPC facility, supported by QMUL Research-IT. http://doi.org/10.5281/zenodo.438045.

## Author contributions

**Conceptualization:** Michael B. Singer, Katerina Michaelides.

**Data curation:** Stuart W. D. Grieve.

**Investigation:** Michael B. Singer, Shiuan-An Chen, Katerina Michaelides.

**Methodology:** Stuart W. D. Grieve, Michael B. Singer, Shiuan-An Chen, Katerina Michaelides.

**Software:** Stuart W. D. Grieve.

**Visualization:** Stuart W. D. Grieve, Shiuan-An Chen.

**Writing – original draft:** Stuart W. D. Grieve.

**Writing – review & editing:** Stuart W. D. Grieve, Michael B. Singer, Shiuan-An Chen, Katerina Michaelides.

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
