## [Decision Letter · Decision Letter 0]

8 Dec 2024

PONE-D-24-28801GDBM: A database of global drainage basin morphologyPLOS ONE

Dear Dr. Grieve,

Thank you for submitting your manuscript to PLOS ONE. After careful consideration, we feel that it has merit but does not fully meet PLOS ONE’s publication criteria as it currently stands. Therefore, we invite you to submit a revised version of the manuscript that addresses the points raised during the review process. Please ensure that your decision is justified on PLOS ONE’s publication criteria and not, for example, on novelty or perceived impact.

We look forward to receiving your revised manuscript.

Kind regards,

Huasheng Huang

Academic Editor

PLOS ONE

Journal Requirements:

2. We note that Figure 1 in your submission contain [map/satellite] images which may be copyrighted. All PLOS content is published under the Creative Commons Attribution License (CC BY 4.0), which means that the manuscript, images, and Supporting Information files will be freely available online, and any third party is permitted to access, download, copy, distribute, and use these materials in any way, even commercially, with proper attribution. For these reasons, we cannot publish previously copyrighted maps or satellite images created using proprietary data, such as Google software (Google Maps, Street View, and Earth). For more information, see our copyright guidelines: http://journals.plos.org/plosone/s/licenses-and-copyright.

Additional Editor Comments:

Please revise point-to-point based on the two reviewers' comments.

Reviewers' comments:

Reviewer's Responses to Questions

**Comments to the Author**

1. Is the manuscript technically sound, and do the data support the conclusions?

Reviewer #1: Yes

Reviewer #2: Partly

2. Has the statistical analysis been performed appropriately and rigorously? 

Reviewer #1: Yes

Reviewer #2: No

3. Have the authors made all data underlying the findings in their manuscript fully available?

Reviewer #1: Yes

Reviewer #2: No

4. Is the manuscript presented in an intelligible fashion and written in standard English?

Reviewer #1: Yes

Reviewer #2: Yes

5. Review Comments to the Author

Reviewer #1: I have thoroughly examined the manuscript titled "GDBM: A database of global drainage basin morphology " The study aims to present a new near-Global dataset of Drainage Basin Morphology, GDBM, which provides morphometric measurements of 254,966 basins based on Shuttle Radar Topography Mission (SRTM) topographic data with Köppen-Geiger climate classifications. The authors have done commendable work; however, before getting accepted, the manuscript needs major revision to strengthen the impact of the research. Kindly see the following comments.

1. Although the abstract claims that the dataset was generated by combining Köppen-Geiger climatic classifications with SRTM topography data, it doesn't explain exactly how this was done. Additionally, channels were classified with "high confidence," however it's not obvious how this was determined.

2. In the abstract, novelty is not given enough attention.

3. The abstract needs a strong conclusion regarding the potential impact and applications of the dataset.

4. Using a drainage threshold of 22.5 km² may still introduce some bias, especially in areas with naturally low drainage network densities (such as deserts). Defend (justify) it.

5. The dataset is intimately linked to estimates of worldwide aridity and Köppen-Geiger climatic zones, providing possibilities to examine river and basin morphology in relation to climate. However, it could be difficult to distinguish morphological changes caused by the environment from those resulting from other causes like geology or human activity.

6. The use of the dataset is restricted by the exclusion of the polar climatic zones (ET and EF). Explain the exclusion.

7. The accuracy of the dataset in capturing dryland hydrological dynamics may be negatively affected by differences in the resolution of climatic zone data and Aridity Index data.

8. While channel extraction from Shuttle Radar Topography Mission (SRTM) data at 30-meter resolution is a well-established approach in geomorphology; however, it has drawbacks compared to modern LiDAR data. Give a reason for using the SRTM data.

9. Descriptions of many variables in the table 2 and 3 are incomplete and demand more details.

Reviewer #2: The manuscript introduces the GDBM, a near-global dataset of drainage basin morphology. It provides morphometric measurements of 254,966 drainage basins and their longest river channels, derived using SRTM topography data combined with Köppen-Geiger climate classifications. However, the current manuscript has several issues that must be addressed before it can be considered for publication.

1. In March 2024, the journal ESSD published a paper titled "A global dataset of the shape of drainage systems." This work used global 90m DEMs to extract 0.67 million drainage basins larger than 50 km² and their longest river channels (from drainage divide to outlet). It also measured river length, sinuosity, basin area, and aspect ratio. The preprint of this article was available online in September 2023. I am puzzled and surprised as to why this highly relevant and important paper has been completely ignored. The authors should carefully compare the differences between the two studies and rewrite the second paragraph of the Introduction to better highlight the innovations of this paper based on He et al., 2024. I think this is this manuscript's most significant flaw.

2. I suggest adding a section to discuss the limitations of the dataset, such as the exclusion of large basins and long rivers due to the division of DEMs.

3. How do the authors verify that the rivers extracted from desert DEMs (e.g., in the Sahara Desert) are real and not artifacts such as dunes?

4. Please provide a global map of the rivers and basins included in this dataset.

5. Providing only CSV files will limit the usability of this study for readers. Why are shapefile formats not included?

6. Line 296: What does "CO2e" refer to?

6. PLOS authors have the option to publish the peer review history of their article (what does this mean?). If published, this will include your full peer review and any attached files.

Reviewer #1: **Yes: **yes

Reviewer #2: No

---

## [Author Response · Author response to Decision Letter 1]

7 Feb 2025

A full response to reviewers was submitted as a pdf.

---

## [Decision Letter · Decision Letter 1]

25 Feb 2025

GDBM: A database of global drainage basin morphology

PONE-D-24-28801R1

Dear Dr. Grieve,

We’re pleased to inform you that your manuscript has been judged scientifically suitable for publication and will be formally accepted for publication once it meets all outstanding technical requirements.

Kind regards,

Huasheng Huang

Academic Editor

PLOS ONE

Additional Editor Comments (optional):

Reviewers' comments:

Reviewer's Responses to Questions

**Comments to the Author**

1. If the authors have adequately addressed your comments raised in a previous round of review and you feel that this manuscript is now acceptable for publication, you may indicate that here to bypass the “Comments to the Author” section, enter your conflict of interest statement in the “Confidential to Editor” section, and submit your "Accept" recommendation.

Reviewer #1: All comments have been addressed

Reviewer #2: All comments have been addressed

2. Is the manuscript technically sound, and do the data support the conclusions?

Reviewer #1: Yes

Reviewer #2: Yes

3. Has the statistical analysis been performed appropriately and rigorously? 

Reviewer #1: Yes

Reviewer #2: Yes

4. Have the authors made all data underlying the findings in their manuscript fully available?

Reviewer #1: Yes

Reviewer #2: Yes

5. Is the manuscript presented in an intelligible fashion and written in standard English?

Reviewer #1: Yes

Reviewer #2: Yes

6. Review Comments to the Author

Reviewer #1: The manuscript has been improved by the authors . Therefore, I recommend acceptance of the manuscript for publication.

Reviewer #2: The authors have adequately addressed my comments and made the necessary revisions. I have no further comments and recommend the paper for publication.

7. PLOS authors have the option to publish the peer review history of their article (what does this mean?). If published, this will include your full peer review and any attached files.

Reviewer #1: No

Reviewer #2: No

---

## [Editor Report · Acceptance letter]

PONE-D-24-28801R1

PLOS ONE

Dear Dr. Grieve,

I'm pleased to inform you that your manuscript has been deemed suitable for publication in PLOS ONE. Congratulations! Your manuscript is now being handed over to our production team.

Kind regards,

on behalf of

Dr. Huasheng Huang

Academic Editor

PLOS ONE